# The Features of Native Gold in Ore-Bearing Breccias with Realgar-Orpiment Cement of the Vorontsovskoe Deposit (Northern Urals, Russia)

**Sergey Y. Stepanov** [1], **Roman S. Palamarchuk** [2,*], **Dmitry A. Varlamov** [3], **Darya V. Kiseleva** [1], **Ludmila N. Sharpyonok** [4], **Radek Škoda** [5] **and Anatoly V. Kasatkin** [6]

1 The Zavaritsky Institute of Geology and Geochemistry UB RAS, Akademika Vonsovskogo Str. 15, 620016 Ekaterinburg, Russia; Stepanov-1@yandex.ru (S.Y.S.); podarenka@mail.ru (D.V.K.)

2 South Urals Federal Research Center of Mineralogy and Geoecology UB RAS, Territory of the Ilmeny State Reserve, 456317 Miass, Russia

3 Institute of Experimental Mineralogy RAS, Academika Osypyana Str. 4, 142432 Chernogolovka, Russia; dima@iem.ac.ru

4 All-Russian Geological Institute Named after A.P. Karpinsky, Sredny Prospect 74, 199106 Saint Petersburg, Russia; lyudmila_sharpenok@vsegei.ru

5 Department of Geological Sciences, Faculty of Science, Masaryk University, 602 00 Brno, Czech Republic; rskoda@sci.muni.cz

6 Fersman Mineralogical Museum of the Russian Academy of Sciences, Leninskiy Prospekt 18/2, 119071 Moscow, Russia; anatoly.kasatkin@gmail.com

* Correspondence: palamarchuk22@yandex.ru; Tel.: +7-(981)-979-9395

**Abstract:** This paper describes native gold in ore-bearing breccias with realgar-orpiment cement from the Vorontsovskoe gold deposit (Northern Urals, Russia). Particular attention is paid to the morphological features of native gold and its relation to other minerals. The latter include both common (orpiment, barite, pyrite, prehnite, realgar) and rare species (Tl and Hg sulfosalts, such as boscardinite, dalnegroite, écrinsite, gillulyite, parapierrotite, routhierite, sicherite, vrbaite, etc.). The general geological and geochemical patterns of the Turyinsk-Auerbakh metallogenic province, including the presence of small non-economic copper porphyry deposits and general trend in change of the composition of native gold (an increase in the fineness of gold from high-temperature skarns to low-temperature realgar-orpiment breccias) confirm that the Vorontsovskoe deposit is an integral part of a large ore-magmatic system genetically associated with the formation of the Auerbakh intrusion.

**Keywords:** native gold; fluid-explosive breccia; Vorontsovskoe gold deposit; Northern Urals; Turyinsk-Auerbakh metallogenic province; Vorontsovsko-Peshchanskaya porphyry system; rare Tl and Hg sulfosalts

## 1. Introduction

The Vorontsovskoe gold deposit is located in Krasnoturyinskiy district of Sverdlovsk Oblast (Northern Urals, Russia), 13 km south of the town of Krasnoturyinsk and approximately 310 km north of Ekaterinburg. It was discovered in 1985 and is currently operated by the Polymetal International PLC. It is located within the Turyinsk-Auerbakh metallogenic province (Figure 1a), which includes a series of medium-size copper-iron-skarn and small-size gold skarn, medium-size lode and quartz stockwork gold deposits with "berezite" (quartz-sericite-carbonate metasomatites) alteration gold deposits. The ore district combines two magmatic-hydrothermal systems [1]: Vorontsovsko-Peshchanskaya and Turyinskaya ($D_{1-2}$). Within the Vorontsovsko-Peshchanskaya system, mainly iron-skarn and gold ore deposits are located. The Turyinskaya system includes a number of medium-size copper-skarn deposits [2] and small-size porphyry copper deposits. By 2003, three economic deposits with calculated reserves were discovered within the Turyinsk-Auerbakh

metallogenic province (Figure 1b): Peshchanskoe iron skarn deposit (approximately 80 million tons of iron); Vorontsovskoe gold deposit (approximately 30 tons of gold, including the off-balance reserves); and the small gold and medium copper Vadimo-Aleksandrovskoe deposit (approximately 6 million tons of copper, including the off-balance reserves and approximately 1 ton of gold) [3].

Some researchers [1] consider the Vorontsovskoe gold deposit to be a peripheral zone of the porphyry copper system. Naumov et al. [4] also classifies this deposit as magmatic-hydrothermal type. Numerous authors [5–8] attribute the Vorontsovskoe gold ore deposit to the Carlin type. Despite such a comprehensive study of the Vorontsovskoe gold deposit, it has not yet been possible to unambiguously resolve the issue of its genesis.

Several types of ores associated with various types of alteration have been identified within the Vorontsovskoe deposit, such as calcareous gold-magnetite-sulfide skarns, quartz-sericite, jasperoids and other alteration types [5,6], supergene alterations [6] as well as gold-pyrite-realgar breccias [9]. We also confirm the assessment of the scale of distribution of various types of ores and their gold content within the Vorontsovskoe deposit and show the leading economic value of gold-ore breccias with orpiment-realgar cement [9,10]. In addition, native gold in this breccia ore type associates with thallium and mercury sulfosalts, that distinguishes the gold mineralization of the Vorontsovskoe deposit from all previously known gold ore localities in Russia. The ore-bearing role of breccias in many deposits is very important [11]. Despite such an important role of the gold breccias of the Vorontsovskoe deposit, the morphological features and composition of their gold have not yet been studied. The aim of our work-detailed characterization of gold from breccias with realgar-orpiment cement.

## 2. Geological Setting

### 2.1. Regional Geological Setting

The Vorontsovskoe gold deposit is located on the eastern slope of the Northern Urals in the eastern part of the Tagil volcanic megazone (Figure 1a). It is located within the North-striking Devonian volcanic belt [5,6]. The Middle Devonian Auerbakh gabbro-diorite-granodiorite intrusion is located in the southern part of the belt [12], and is associated with the formation of the Turyinsk-Auerbakh metallogenic province [1].

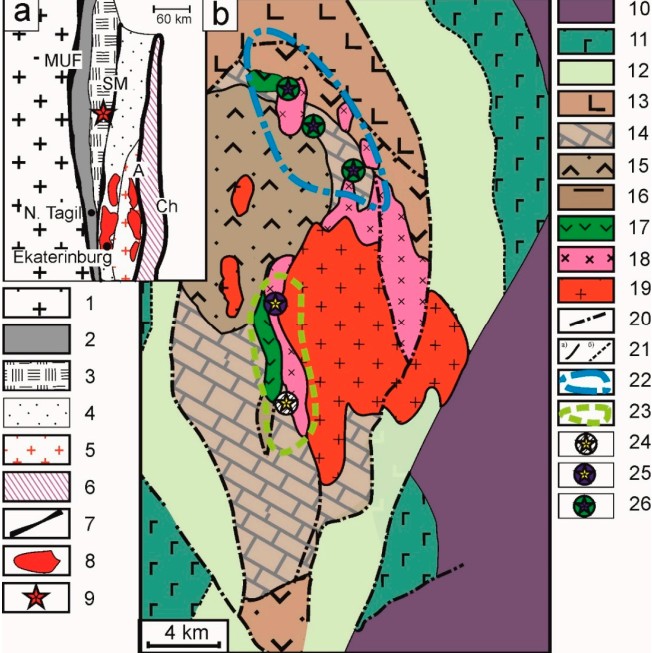

**Figure 1.** The position of the Auerbakh intrusion in the structures of the Urals (a) and geological scheme of the structure of the Turyinsk-Auerbakh metallogenic province (b). (**a**): 1—the passive margin

of the western slope of the Urals; 2—the platinum belt of Tagil megazone; 3—Tagil volcanic zone; the zones of the active continental margin: 4—Devonian-Carboniferous; 5—continental; 6—northern; 7—tectonic sutures separating large structures of the Urals: the Main Uralian Fault (MUF), Serovsko-Mauksky (SM), Alapaevsky (A), Chelyabinsky (Ch). Plutonic massifs: 8—granite; 9—Auerbakh gabbro-diorite-granite (after [13]). (**b**): 10—Ordovician rocks of ophiolitic association; 11—basalt-rhyolite-plagiogranite association, Ordovician; 12—Silurian trachybasalt-trachyte; Devonian formations: 13—volcanic-sedimentary rocks with intercalations of andesites, andesite-basalts and limestones; 14—reef limestone; 15—volcanomictic rocks with tuff horizons of andesites and andesidacites; 16—andesibasalts; 17—extrusive andesites. Igneous rocks of the Auerbakh Intrusion: 18—porphyritic diorites and gabbro-diorites; 19—quartz diorites, granodiorites and granites; other designations: 20—faults; 21—rock contacts: a) sharp; b) gradual; 22—Turyinskaya ore-magmatic system; 23—Vorontsovsko-Peshchanskaya ore-magmatic system; Deposits: 24—Vorontsovskoe gold ore deposit; 25—Peshchanskoe iron ore deposit; 26—copper-skarn deposit (after [1]).

The Vorontsovsko-Peshchanskaya hydrothermal system is located at the southwestern outer contact of the Auerbakh intrusion [1]. The Vorontsovskoe deposit is located in the southern part of this system, at a distance of 400–500 m from the southwestern exocontact of the Auerbakh massif (Figure 1b) [7]. The volcanic-sedimentary rocks hosting the deposit form a monocline that gently dips to the west [6]. The sedimentary sequence includes limestones and layers of tuffites and siltstones with a thickness of approximately 1 km. Usually, the limestone is metamorphosed to marbles. This sequence is conformably overlain subsequently by volcanic-sedimentary and volcanic rocks, such as tuffaceous siltstone, tuffite and tuff. Coarse clastic breccias with tuffaceous cement are common at the contact of all these rocks with limestones within the entire Turyinsk-Auerbakh metallogenic province [14].

### 2.2. Local Geological Setting

A quarry in the Vorontsovskoe deposit has exposed a wedge body of volcano-sedimentary rocks with a predominance of tuffs of medium composition and tuffstones (Figure 2a). The western part of this body is bounded by a large tectonic fault (Figure 2b). The bulk of ore-bearing breccia, including realgar-orpiment cement, is located at the contact of the body of volcano-sedimentary rocks with limestones. The gold ore body has the form of a torch that expands outward the top [15]. Within this body, gold mineralization is localized mainly in breccias. Part of the gold mineralization is associated with metasomatically sericite-altered volcanic-sedimentary rocks.

Two stages of ore breccia formation were previously identified earlier [9]. The breccias of the first stage are quite widespread. Limestone fragments predominate in this type of breccias. These fragments are embedded in a matrix consisting of small fragments of volcanic-sedimentary rocks dominated by andesite [16,17]. Pyrite grains with an average size of 0.2 mm are widely distributed in the cement. The breccias of the second stage form a pod-like body of irregular shape (Figure 2c). Limestones, volcanogenic sedimentary rocks and siltstones are found as fragments in breccias of the second stage. First-stage breccias are also found in the wreckage. Realgar and orpiment are widely distributed in breccia cement of the second stage. Barite, quartz and calcite make up a significant part of the cement. Native gold and rare Hg and Tl sulfosalts are common in breccia cement of the second stage. The highest content of realgar and orpiment is typical for the central parts of pod-like bodies of the second stage breccias.

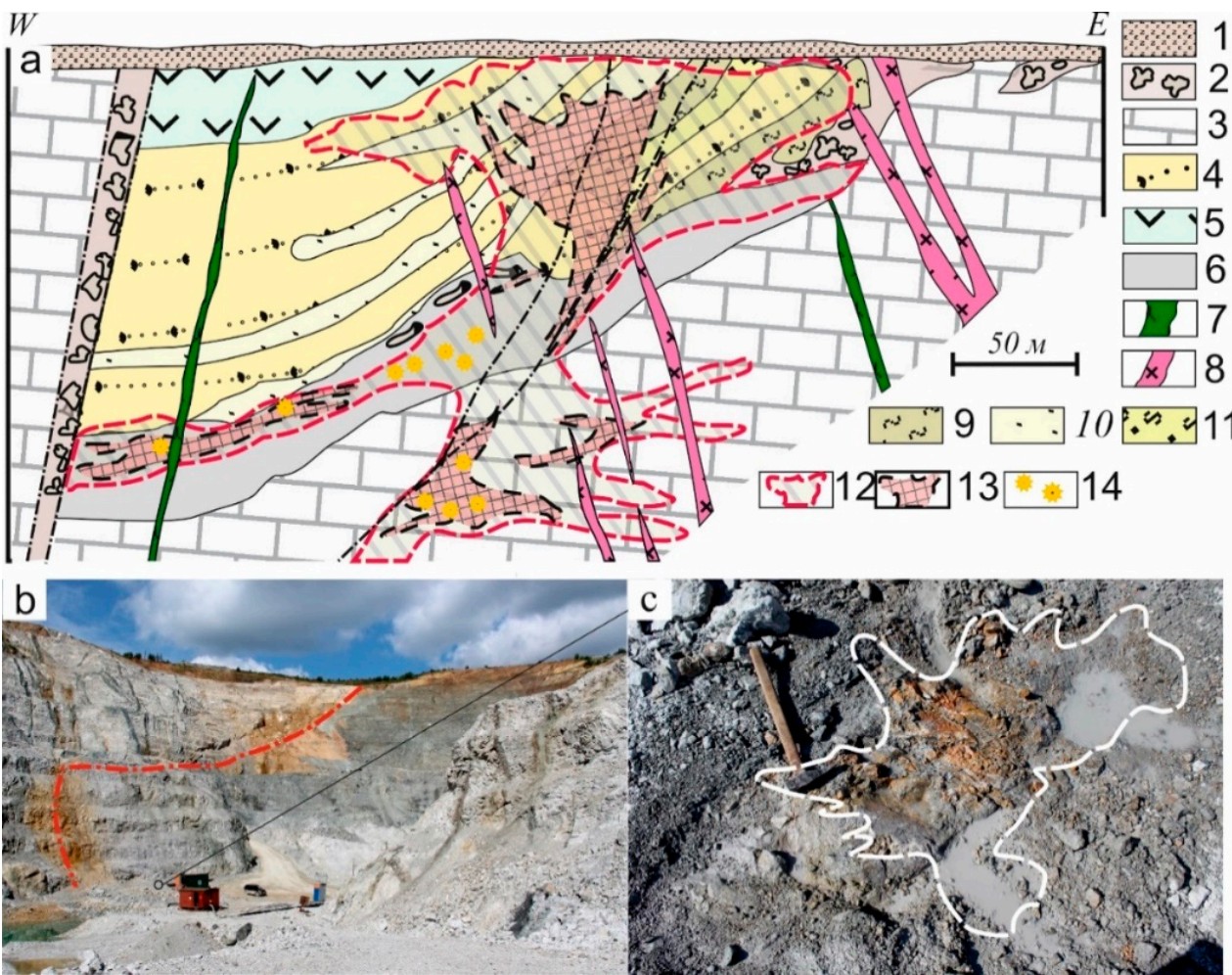

**Figure 2.** (**a**) Schematic section of the Vorontsovskoe deposit [15–17]; (**b**) the photograph of the northern wall of the quarry with a tectonic contact between marmorized limestone and a layer of volcanic-sedimentary rocks (red dotted line); (**c**) a small body of fluid-explosive breccias with realgar-orpiment cement (sample 2016/2), exposed in one of the quarry layers. The white dotted line shows the boundary of the body. 1—Neogene-Quaternary cover deposits; 2—karst formations; Devonian formations: 3—limestones; 4—tuff aleurolites, tuffstones, tuff-conglomerates; 5—andesites, tuffs and lava breccias; 6—breccia of the 1st stage of breccia formation; 7—lamprophyre dikes; 8—dikes of diorite porphyrites. Metasomatites: 9—quartz-sericite, 10—quartz-sericite-albite, 11—berezite-listvenites, chlorite-sericites. 12—ore bodies with run-of-mine grades of gold content; 13—enriched ore pillars; 14—manifestation areas of realgar-orpiment mineralization.

## 3. Materials and Methods

### 3.1. Sample Collection and Preparation

A total of 173 ore samples were taken within the ore body to study the distribution of gold. Of these, 58 specimens were from ore breccias with realgar-orpiment cement. These samples were collected in 2016–2019 at the Northern quarry of the Vorontsovskoe deposit at horizons from –40 to +35 (meters above sea level) from 2016 to 2019 (Figure 3). The weight of ore samples ranged from 0.2 to 0.4 kg.

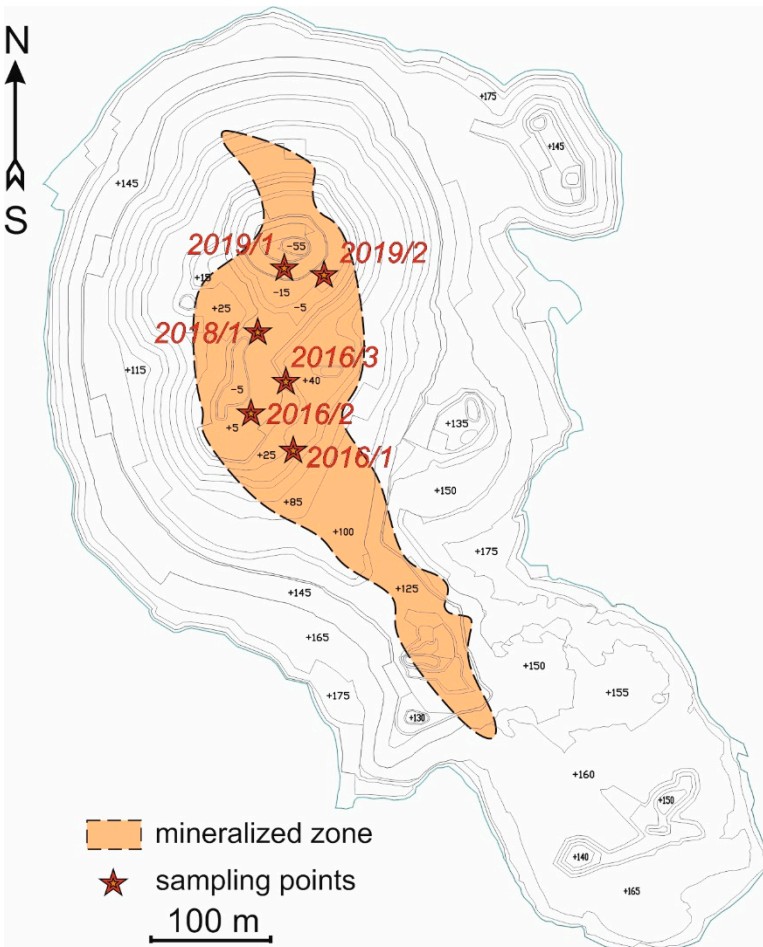

**Figure 3.** Location of sampling at the Northern quarry of Vorontsovsky deposit. First, digit means the year of collecting, second—the number of sampling point.

To obtain native gold concentrates, two large samples weighing 35 kg were taken from areas richest in realgar and orpiment breccia fragments. The samples were crushed to a fraction of less than 1 mm and enriched with a KR-400 centrifugal concentrator. Grains and crystals of native gold were manually selected using a binocular microscope. Their surface structure details were studied by scanning electron microscopy. Then, the grains of native gold were mounted in polished sections in order to study their internal structure and determine their chemical composition.

*3.2. Analytical Methods*

Polished and thin sections were made for microscopic examination. After a preliminary description using an optical microscope under transmitted and reflected light, detailed studies of thin sections with native gold using a scanning electron microscope were carried out. The chemical compositions of minerals were identified by an electron microprobe using both energy and wavelength dispersive spectrometers.

A preliminary semi-quantitative analysis of the chemical composition was performed in the Fersman Mineralogical Museum of the Russian Academy of Sciences (Moscow, Russia) using a CamScan 4D scanning electron microscope (CamScan Electron Optics, Ltd, Cambridge, UK) and in the Institute of Experimental Mineralogy RAS (Chernogolovka, Russia) using a CamScan MV2300 scanning electron microscope. In both cases, INCA Energy 350 energy dispersive spectrometers (Tescan, Brno, Czech Republic) were used under the following operational conditions: accelerating voltage—20 kV, probe current—5 nA on metallic cobalt, working distance 25 mm, spectra accumulation time—70 s, spot size 5 μm.

Further study of the mineral chemical composition was carried out in the joint laboratory of electron microscopy and microanalysis of the Department of Geological Sciences of the Masaryk University and the Czech Geological Survey (Brno, Czech Republic) using a Cameca SX 100 wave dispersive electron probe microanalyzer (Cameca, Paris, France) and in the laboratory of the Department of Mineralogy of the Geological Faculty of Moscow State University (Moscow, Russia) using the Camebax SX 50 microanalyzer (Cameca, Paris, France). In the first case, the following operational conditions were applied: accelerating voltage—25 kV, probe current—20 nA, probe diameter—1 μm; reference materials (natural): Fe—$FeS_2$; Cu—Cu metal; Zn— ZnS; Ag—Ag metal; Hg—HgTe; Tl—Tl (Br, I); Pb and Se—PbSe; As—pararammelsbergite; Sb—Sb; S—chalcopyrite. In the second case, the operational conditions were as follows: accelerating voltage—20 kV, probe current—30 nA, probe diameter—1 μm. The following reference materials were used (natural): Zn—ZnS, As—CoAsS, S, Fe—FeS (troilite), Ag—$Ag_2Te$, Cu, Sb—$CuSbS_2$, Hg—HgTe, Tl—$TlSbSe_2$, Pb—PbS.

## 4. Results

### 4.1. Mineral Composition of the Breccias

The gold-bearing breccias of the second brecciation event consist of marmorized limestone, tuffstones, tuffites and andesite tuffs fragments and contain fragments of individual grains of minerals from these rocks embedded in the hydrothermal cement (manganoan calcite, prehnite, orthoclase and other). Rock fragments have an angular shape (Figure 4) and vary in size from one millimeter to several centimeters. The quantitative ratio of the matrix and lithoclasts in breccias is not constant and varies from 15 to 75%. The breccias are altered to varying degrees. In the breccia matrix, thin (less than 1 mm) fragments of the main rock-forming minerals are widespread: chlorite (clinochlore and chamosite), amphibole (magnesio-ferri-hornblende, tremolite, pargasite), scapolite, quartz, feldspars (orthoclase, microcline, albite) and calcite. However, newly formed minerals predominate, forming the cement of ore breccias.

The most common hydrothermal gangue minerals of breccia cement are manganoan calcite, prehnite, orthoclase var. hyalophane, fluorapatite and barite. Prehnite forms small, idiomorphic prismatic crystals or granular masses cemented by ore minerals—pyrite, realgar, stibnite, aktashite, boscardinite, parapierrotite, routhierite and chabournéite (Figure 5a,b,d,e). Orthoclase var. hyalophane forms intergrowths with prehnite granular aggregates (Figure 5b), or, more rarely, single prismatic crystals. Fluorapatite is found as rare prismatic crystals included in the aggregates of other minerals (Figure 5b). Barite and manganoan calcite form only fine-grained masses (Figure 5c). Gangue minerals are intergrown mainly with realgar (Figure 5b,d,e) and less often with orpiment.

Stibnite (Figure 5d), orpiment, realgar (Figure 5b,d,e) and pyrite (Figure 5a,b,d,e) prevail among the ore minerals in the breccia cement. Pyrite forms idiomorphic pentagonal dodecahedra and often contains up to 4 wt.% As in its structure. Rare Tl- and Hg-bearing sulfosalts, sulfides and tellurides are found among the ore cement minerals and include aktashite, bernardite, boscardinite, weissbergite, vrbaite, gillulyite, dalnegroite, sicherite, imhofite, coloradoite, christite, laffittite, lorándite, parapierrotite, picotpaulite, rebulite, routhierite, philrothite, hutchinsonite, chabournéite, écrinsite, etc. (full list of minerals identified at Vorontsovskoe deposit including rare Tl-Hg-bearing ones is given by [16–18]). These ore minerals were formed later than most gangue minerals. Thus, aktashite and boscardinite cement prehnite crystals (Figure 5a,b), parapierrotite replaces stibnite (Figure 5d), routhierite cements individual prehnite and diopside crystals (Figure 5d) and chabournéite fills veinlets intersecting silicates and realgar (Figure 5c,e). The chemical composition of selected rare Tl and Hg minerals mentioned above is given in Table 1.

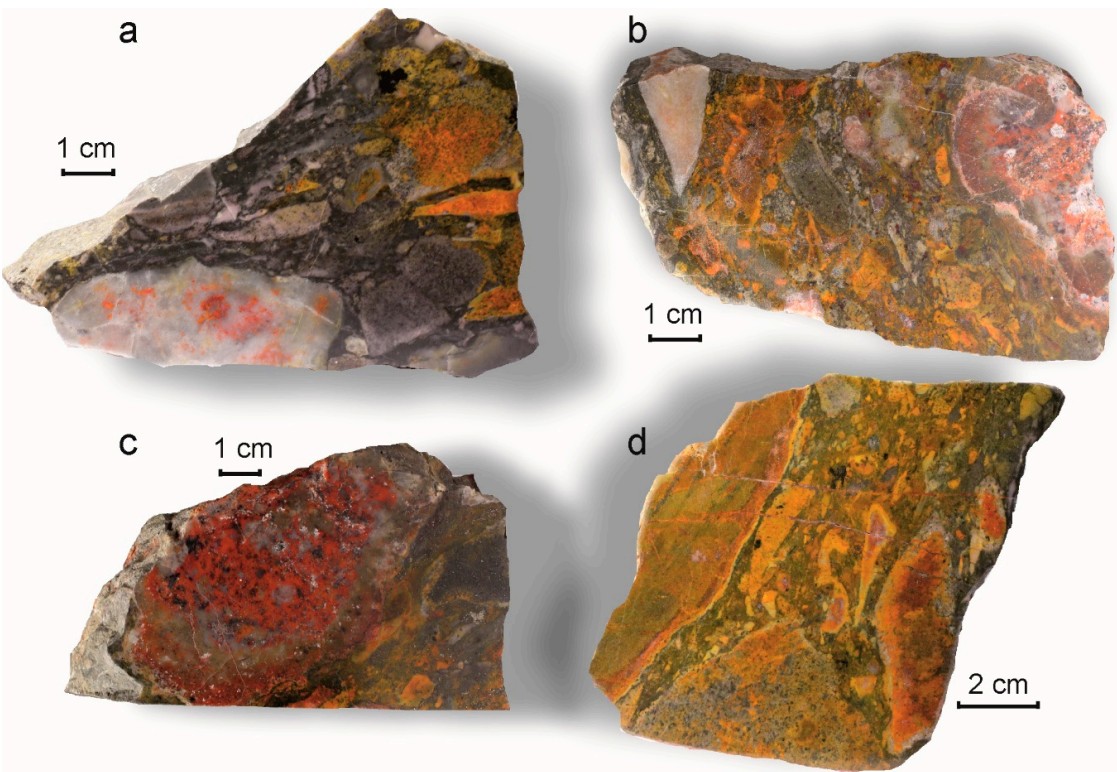

**Figure 4.** The varieties of gold ore breccias in the Vorontsovskoe deposit: (**a**) with a predominance of tuffstone and a small amount of orpiment-realgar cement in the fragments; (**b**) with a relatively equal amount of fragments of tuffstones (strongly altered) and marmorized limestones with a moderate amount of orpiment-realgar cement; (**c**) with equal amounts of fragments of tuffstones and marmorized limestones and a prevalence of realgar cement (black points are the accumulations of Tl and Hg sulfides and sulfosalts); (**d**) with a prevalence of metasomatically transformed tuffstones with a significant amount of orpiment-realgar cement.

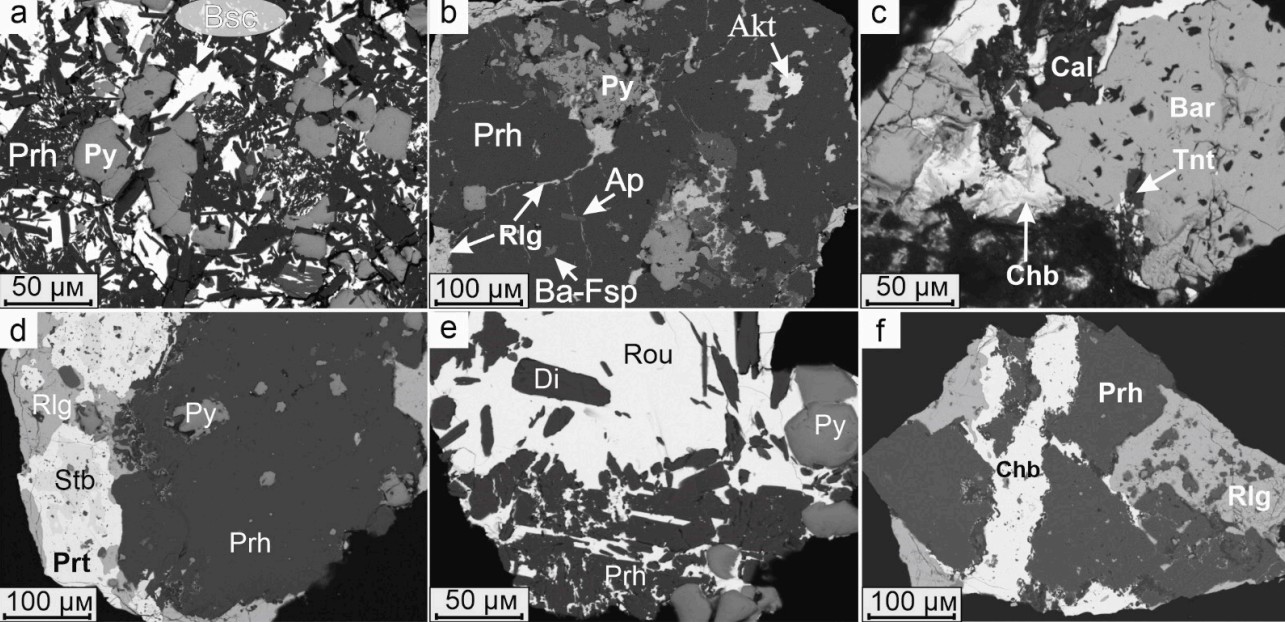

**Figure 5.** Gangue minerals from gold ore breccia cement. (**a**) The relationship of pyrite (Py) with boscardinite (Bsc) and prehnite (Prh); (**b**) aktashite (Akt) grain in hyalophane (Ba-Fsp); (**c**) the relationship of chabournéite (Chb) with calcite (Cal) and barite (Bar); (**d**) parapierrotite (Prt) and stibnite (Stb) with large prehnite grain; (**e**) routhierite (Rou) cements individual prehnite and diopside (Di) crystals; (**f**) prehnite in realgar-chabournéite cement. Ap—fluorapatite, Tnt—tennantite.

**Table 1.** The chemical compositions of Tl and Hg sulfosalts from the cement of gold ore breccias, wt. %.

| № | Fe | Cu | Ag | Hg | Tl | Pb | As | Sb | S | Total | Mineral |
|---|---|---|---|---|---|---|---|---|---|---|---|
| 1 | 1.58 | 25.48 | - | 24.29 | - | - | 19.80 | 0.49 | 25.45 | 99.85 | Aktashite |
| 2 | - | - | - | - | 21.40 | 0.13 | 25.32 | 25.39 | 27.98 | 100.22 | Bernardite |
| 3 | - | - | 2.01 | - | 12.60 | 17.82 | 12.19 | 30.86 | 24.22 | 99.70 | Boscardinite |
| 4 | - | - | - | - | 53.46 | - | 5.17 | 24.73 | 16.80 | 100.16 | Weissbergite |
| 5 | - | - | - | 20.42 | 28.52 | - | 20.61 | 8.57 | 22.32 | 100.44 | Vrbaite |
| 6 | - | - | - | - | 29.49 | 0.19 | 40.02 | 1.73 | 29.52 | 100.95 | Gillulyite |
| 7 | - | - | - | - | 19.20 | 9.99 | 21.04 | 23.43 | 25.82 | 99.48 | Dalnegroite |
| 8 | 0.10 | - | 23.83 | - | 22.63 | 0.22 | 16.71 | 13.69 | 21.61 | 98.79 | Sicherite |
| 9 | - | - | - | - | 36.30 | - | 30.02 | 8.43 | 25.37 | 100.12 | Imhofite |
| 10 | - | - | - | 34.50 | 35.73 | - | 12.62 | - | 16.40 | 99.25 | Christite |
| 11 | - | - | 23.10 | 40.90 | - | - | 15.95 | - | 20.16 | 100.11 | Laffittite |
| 12 | - | - | - | - | 58.49 | 0.42 | 20.57 | 0.45 | 19.37 | 99.30 | Lorándite |
| 13 | - | - | - | - | 20.10 | - | 8.61 | 46.08 | 25.17 | 99.96 | Parapierrotite |
| 14 | 27.21 | - | - | - | 49.29 | - | - | - | 23.86 | 100.36 | Picotpaulite |
| 15 | - | - | - | - | 34.80 | 0.36 | 22.98 | 16.82 | 22.23 | 99.19 | Rebulite |
| 16 | - | 5.92 | 0.25 | 38.87 | 19.21 | - | 13.97 | 1.72 | 19.03 | 99.48 [1] | Routhierite |
| 17 | - | - | - | - | 33.17 | 0.59 | 34.40 | 3.94 | 27.06 | 99.16 | Philrothite |
| 18 | - | - | - | - | 19.14 | 18.68 | 31.25 | 4.73 | 26.54 | 100.34 | Hutchinsonite |
| 19 | - | - | - | - | 17.47 | 10.96 | 13.86 | 32.89 | 24.85 | 100.03 | Chabournéite |
| 20 | 0.25 | - | 1.55 | - | 8.72 | 26.04 | 17.84 | 20.78 | 24.28 | 99.62 [2] | Ecrinsite |

| № | APFU | Empirical Formulae | Mineral |
|---|---|---|---|
| 1 | 25 | $Cu_{6.06}(Hg_{1.83}Zn_{0.64}Fe_{0.43})_{\Sigma 2.90}(As_{3.99}Sb_{0.06})_{\Sigma 4.05}S_{11.99}$ | Aktashite |
| 2 | 14 | $Tl_{0.96}Pb_{0.01}(As_{3.10}Sb_{1.92})_{\Sigma 5.02}S_{8.01}$ | Bernardite |
| 3 | 64 | $Ag_{0.89}Tl_{2.95}Pb_{4.11}(Sb_{12.12}As_{7.78})_{\Sigma 19.90}S_{36.14}$ | Boscardinite |
| 4 | 4 | $Tl_{0.99}(Sb_{0.77}As_{0.26})_{\Sigma 1.03}S_{1.98}$ | Weissbergite |
| 5 | 37 | $Hg_{2.94}Tl_{4.02}As_{7.93}Sb_{2.03}S_{20.08}$ | Vrbaite |
| 6 | 22.8 | $Tl_{2.04}Pb_{0.01}(As_{7.54}Sb_{0.21})_{\Sigma 7.75}S_{13.00}$ | Gillulyite |
| 7 | 60 | $Tl_{3.97}Pb_{2.04}(As_{11.86}Sb_{8.13})_{\Sigma 19.99}S_{34.01}$ | Dalnegroite |
| 8 | 12 | $Tl_{0.99}Pb_{0.01}(Ag_{1.97}Fe_{0.02})_{\Sigma 1.99}(As_{1.99}Sb_{1.00})_{\Sigma 2.99}S_{6.02}$ | Sicherite |
| 9 | 47.2 | $Tl_{5.83}(As_{13.14}Sb_{2.27})_{\Sigma 15.41}S_{25.96}$ | Imhofite |
| 10 | 6 | $Tl_{1.02}Hg_{1.01}As_{0.98}S_{2.99}$ | Christite |
| 11 | 6 | $Ag_{1.02}Hg_{0.97}As_{1.01}S_{2.99}$ | Laffittite |
| 12 | 4 | $Tl_{0.98}Pb_{0.01}(As_{0.94}Sb_{0.01})_{\Sigma 0.95}S_{2.06}$ | Lorándite |
| 13 | 14 | $Tl_{1.00}(Sb_{3.85}As_{1.17})_{\Sigma 5.02}S_{7.98}$ | Parapierrotite |
| 14 | 6 | $Tl_{0.98}Fe_{1.99}S_{3.03}$ | Picotpaulite |
| 15 | 40 | $Tl_{4.96}Pb_{0.05}(As_{8.94}Sb_{4.03})_{\Sigma 12.97}S_{22.02}$ | Rebulite |
| 16 | 12 | $(Cu_{0.94}Ag_{0.02})_{\Sigma 0.96}(Hg_{1.96}Zn_{0.08})_{\Sigma 2.04}Tl_{0.95}(As_{1.89}Sb_{0.14})_{\Sigma 2.03}S_{6.01}$ | Routhierite |
| 17 | 9 | $Tl_{0.97}Pb_{0.02}(As_{2.75}Sb_{0.19})_{\Sigma 2.94}S_{5.06}$ | Philrothite |
| 18 | 16 | $Tl_{1.02}Pb_{0.98}(As_{4.55}Sb_{0.42})_{\Sigma 4.97}S_{9.03}$ | Hutchinsonite |
| 19 | 60 | $Tl_{3.75}Pb_{2.32}(Sb_{11.84}As_{8.11})_{\Sigma 19.95}S_{33.98}$ | Chabournéite |
| 20 | 64 | $Ag_{0.68}Fe_{0.21}Tl_{2.01}Pb_{5.93}(As_{11.24}Sb_{8.06})_{\Sigma 19.30}(S_{35.76}Se_{0.10})_{\Sigma 35.86}$ | Ecrinsite |

Notes: [1] the total includes 0.51 wt.% of Zn; [2] the total includes 0.16 wt.% of Se.

### 4.2. Morphological Features of Native Gold in Breccias

Native gold in the breccias occurs both in cement (Figure 6a) and in the limestone's fragments. The analysis of ore concentrates obtained during the gravitational concentration of gold breccias with realgar-orpiment cement has made it possible to establish that the grains of 0.1–0.25 mm prevail among the gravitationally enriched aggregate of gold. However, the study of thin sections has revealed the predominance of gold grains less than 0.1 mm in size. Thus, most of the native gold grains (75% of all grains) are 20–80 μm in size. Large grains of native gold (0.2 to 0.8 mm) are quite rare. These grains occur as complex intergrowths of gold with realgar, orpiment and calcite (Figure 6b,c).

The study of thin sections from breccias allowed us to confirm the conclusion that most of the native gold is located directly in the breccias' cement. Gold grains intergrow with orpiment and realgar (Figure 7a,b). Gold grains with prismatic cross-sections are extremely rare and are located directly in the fragments of marmorized limestones

(Figure 7c). Gold crystals are relatively rare among the gold grains from the ore concentrates too (Figure 8a). They are characterized by a complex faceting with a combination of octahedron, pentagonal dodecahedron and cubic faces, which leads to the occurrence of spherical shape of the crystals.

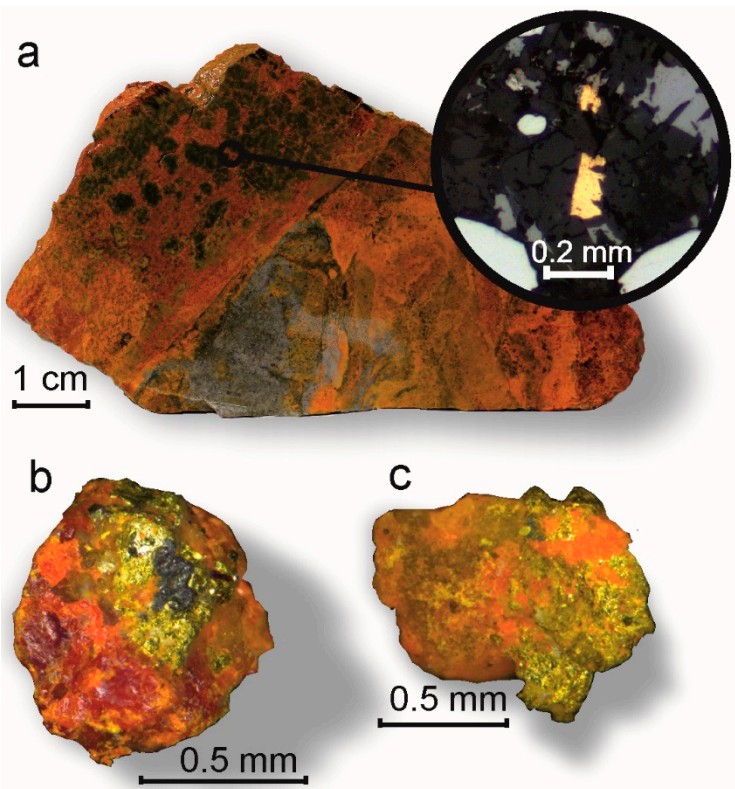

**Figure 6.** Native gold occurrences in ore breccias: (**a**) gold realgar-orpiment breccia with small pyrite grains and native gold, (**b**) native gold with realgar, (**c**) native gold with orpiment and calcite.

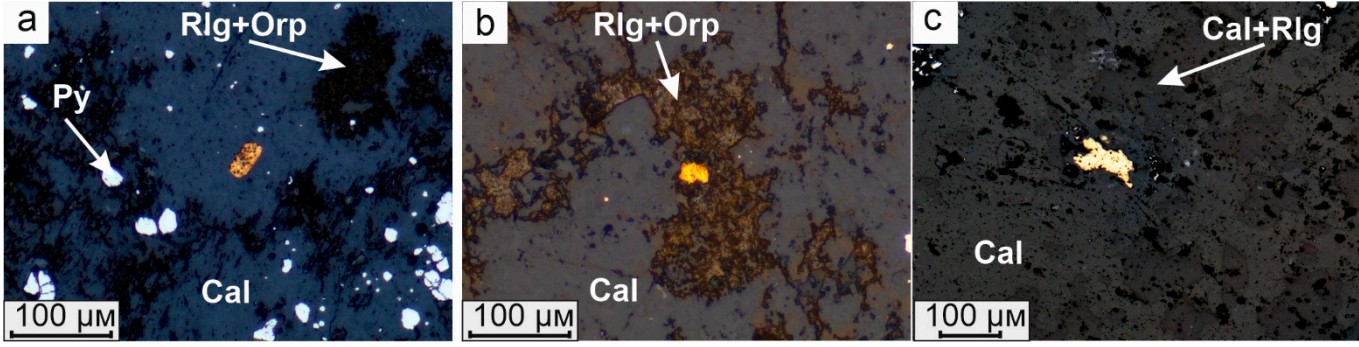

**Figure 7.** Optical photo of the native gold in realgar-orpiment breccias of the Vorontsovskoe deposit: (**a**) gold grain and pyrite (Py) in calcite; (**b**) the grain of gold in realgar-orpiment (Rlg + Orp) breccia cement; (**c**) gold in brecciated calcite marble; Cal + Rlg—the fine-crystalline calcite and realgar.

Gold grains cementing other minerals are most widely distributed (Figure 8b–e). Often, native gold in such aggregates intergrows with realgar, pyrite and calcite.

Native gold contains numerous mineral inclusions. Small crystals of pyrite, prehnite, quartz and calcite are directly enclosed in the native gold. Polymineral inclusions, consisting of calcite, barite and dalnegroite, are found in gold aggregates as well (Figure 9a). Polymineral inclusions formed by prehnite, barite and routhierite are also detected (Figure 9b). Hyalophane grains are present as inclusions in gold in a very limited quantity. The above-listed minerals can be found not only in the form of inclusions in gold but also as in-

tergrowths with gold. Among the rare minerals intergrown with gold, we also note coloradoite, arsenolite (Figure 9c) and parapierrotite (Figure 9d). Arsenolite is the most recent mineral formed as a result of supergene processes.

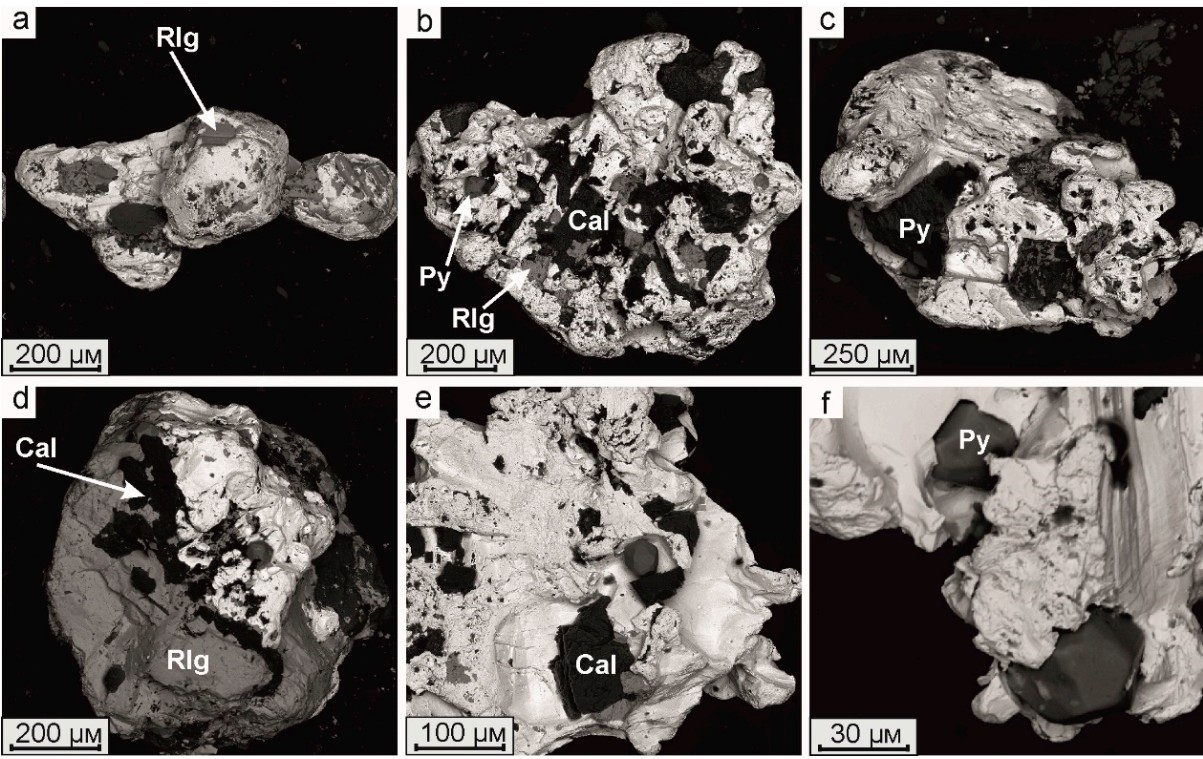

**Figure 8.** BSE-photo of the morphology of native gold grains and aggregates from breccias with orpiment-realgar cement (explained in the text). (**a**) gold crystal; (**b**) gold with mineral inclusions of calcite (Cal), pyrite (Py) and realgar (Rlg); (**c**) native gold intergrows with pyrite; (**d**) native gold intergrows with realgar; (**e**) native gold intergrows with calcite; (**f**) pyrite crystals in native gold.

**Table 2.** The chemical composition of minerals intergrown with native gold, wt. %.

| № | S | Fe | Cu | As | Sb | Hg | Tl | Pb | Total | Mineral |
|---|---|---|---|---|---|---|---|---|---|---|
| 1 | 27.41 | – | – | 31.46 | 10.15 | – | 20.87 | 10.36 | 100.25 | Dalnegroite |
| 2 | 27.45 | – | – | 31.50 | 10.31 | – | 20.80 | 10.37 | 100.43 | Dalnegroite |
| 3 | 27.59 | – | – | 31.63 | 10.52 | – | 20.55 | 10.45 | 100.74 | Dalnegroite |
| 4 [1] | 19.26 | 0.09 | 6.18 | 14.55 | 0.45 | 36.88 | 21.33 | – | 99.86 | Routhierite |
| 5 [2] | 19.28 | 0.12 | 6.23 | 14.65 | 0.53 | 36.79 | 21.18 | – | 99.87 | Routhierite |
| 6 [3] | – | – | – | – | – | 60.53 | – | – | 100.48 | Coloradoite |
| 7 | 51.22 | 43.70 | 0.10 | 3.93 | – | – | – | – | 98.95 | Pyrite |
| 8 | 24.66 | – | – | 4.13 | 51.12 | – | 20.40 | – | 101.31 | Parapierrotite |

| № | APFU | Empirical Formulae | Mineral |
|---|---|---|---|
| 1 | 60 | $Tl_{4.06}Pb_{1.99}(As_{16.68}Sb_{3.31})_{\Sigma19.99}S_{33.96}$ | Dalnegroite |
| 2 | 60 | $Tl_{4.04}Pb_{1.98}(As_{16.67}Sb_{3.36})_{\Sigma20.03}S_{33.95}$ | Dalnegroite |
| 3 | 60 | $Tl_{3.97}Pb_{1.99}(As_{16.66}Sb_{3.41})_{\Sigma20.07}S_{33.97}$ | Dalnegroite |
| 4 | 12 | $Cu_{0.97}(Hg_{1.83}Zn_{0.17}Fe_{0.02})_{\Sigma2.02}Tl_{1.04}(As_{1.94}Sb_{0.04})_{\Sigma1.98}S_{5.99}$ | Routhierite |
| 5 | 12 | $Cu_{0.98}(Hg_{1.83}Zn_{0.17}Fe_{0.02})_{\Sigma2.02}Tl_{1.03}(As_{1.95}Sb_{0.04})_{\Sigma1.99}S_{5.99}$ | Routhierite |
| 6 | 2 | $Hg_{0.98}Te_{1.02}$ | Coloradoite |
| 7 | 3 | $Fe_{0.96}As_{0.06}S_{1.97}$ | Pyrite |
| 8 | 14 | $Tl_{1.04}(Sb_{4.37}As_{0.57})_{\Sigma4.94}S_{8.01}$ | Parapierrotite |

Notes: the points of analyses are given in Figure 9, [1] the total includes 1.12 wt.% of Zn; [2] the total includes 1.09 wt.% of Zn; [3] the total includes 39.95 wt.% of Te. Dash is element content below detection limits.

**Table 3.** The chemical composition of native gold from breccias with realgar-orpiment cement, wt. %. (fineness in ‰).

| № | Cu | Ag | Au | Hg | Total | Empirical Formulae | Fineness |
|---|---|---|---|---|---|---|---|
| 1 | – | 19.55 | 80.49 | – | 100.04 | $Au_{0.69}Ag_{0.31}$ | 804.6 |
| 2 | – | 16.30 | 83.32 | – | 99.62 | $Au_{0.74}Ag_{0.26}$ | 836.4 |
| 3 | – | 12.38 | 86.38 | 0.46 | 99.22 | $Au_{0.79}Ag_{0.21}$ | 870.6 |
| 4 | – | 11.18 | 86.84 | 0.62 | 98.64 | $Au_{0.80}Ag_{0.19}Hg_{0.01}$ | 880.4 |
| 5 | – | 0.98 | 99.02 | – | 100.00 | $Au_{0.98}Ag_{0.21}$ | 990.2 |
| 6 | – | 1.10 | 99.44 | – | 100.54 | $Au_{0.98}Ag_{0.21}$ | 989.1 |
| 7 | – | 3.40 | 97.41 | 0.13 | 100.94 | $Au_{0.94}Ag_{0.21}$ | 965.0 |
| 8 | – | 0.16 | 99.08 | – | 99.24 | $Au_{1.00}$ | 998.4 |
| 9 | – | 5.43 | 94.38 | 1.05 | 100.86 | $Au_{0.90}Ag_{0.09}Hg_{0.01}$ | 935.8 |
| 10 | 0.13 | 6.60 | 92.70 | 1.16 | 100.59 | $Au_{0.88}Ag_{0.11}Hg_{0.01}$ | 921.6 |
| 11 | – | 5.71 | 92.92 | 0.68 | 99.31 | $Au_{0.89}Ag_{0.10}Hg_{0.01}$ | 935.7 |
| 12 | – | 2.92 | 98.72 | 0.26 | 101.90 | $Au_{0.95}Ag_{0.05}$ | 968.8 |
| 13 | 0.11 | 0.24 | 98.96 | 0.36 | 99.67 | $Au_{0.99}(Ag,Hg,Cu)_{0.01}$ | 992.9 |
| 14 | – | 0.26 | 99.01 | 0.57 | 99.84 | $Au_{0.99}Hg_{0.01}$ | 991.7 |
| 15 | – | – | 99.41 | – | 99.41 | $Au_{1.00}$ | 1000.0 |
| 16 | 0.23 | 0.20 | 99.42 | 0.48 | 100.33 | $Au_{0.99}Cu_{0.01}$ | 990.9 |
| 17 | – | – | 100.13 | 0.59 | 100.72 | $Au_{0.99}Hg_{0.01}$ | 994.1 |
| 18 | 0.11 | 0.24 | 98.96 | 0.36 | 100.69 | $Au_{0.99}Hg_{0.01}$ | 992.9 |
| 19 | 0.07 | 7.83 | 89.66 | 1.85 | 99.41 | $Au_{0.85}Ag_{0.13}Hg_{0.02}$ | 901.9 |
| 20 | 0.05 | 3.73 | 96.86 | – | 100.64 | $Au_{0.93}Ag_{0.07}$ | 962.4 |

Notes: the points of analyses of native gold No. 9–16 are given in Figure 9, No. 1–8, 17–20 are given in Figure 10. Dash is element content below detection limits.

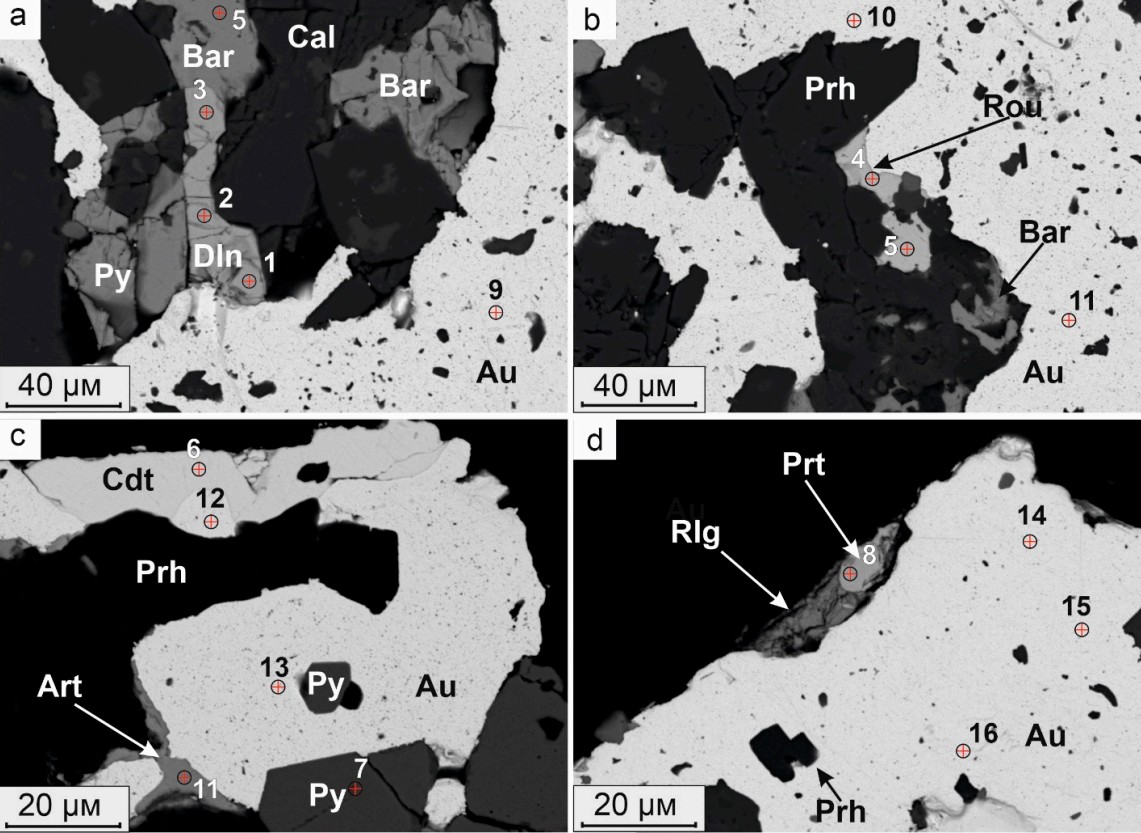

**Figure 9.** BSE-photo of the minerals intergrown with gold from realgar-orpiment cement breccia. Analyses corresponding to 1–8 point numbers are given in Table 2; the analyses of native gold No. 9–16 are given in Table 3. (**a**) Polymineral inclusions, consisting of calcite (Cal), barite (Bar), pyrite (Py) and dalnegroite (Dln); (**b**) polymineral inclusions formed by prehnite (Prh), barite and routhierite (Rou); (**c**) coloradoite (Cdt) and arsenolite (Art) intergrown with gold; (**d**) parapierrotite (Prt), realgar (Rlg) and prehnite crystals in native gold.

In general, the combination of minerals enclosed by or intergrown with the native gold (Table 2) is similar to the mineral composition of ore-bearing breccia cement. The nature of the relationship of native gold with other minerals indicates that its formation was simultaneous with that of most of the sulfide and sulfosalt grains. The presence of calcite, prehnite, hyalophane and other minerals as inclusions in both gold and sulfides and sulfosalts indicates their earlier formation. Thus, the native gold and most of the sulfides and sulfosalts in the breccias' cement should be attributed to the same assemblage, that has formed at the final stage of the formation of the realgar-orpiment breccia cement.

### 4.3. Chemical Composition of Native Gold

Native gold from the ore-bearing breccia cement of the Vorontsovskoe deposit is characterized by a homogeneous internal structure (Figure 10). It is divided into two groups by its chemical composition: 1) high-purity gold with Au content from 95 to 100 wt. % and 2) silver-containing gold with Au content from 80 to 95 wt. % (Table 3; Figure 11). Elevated silver concentrations in native gold are accompanied by elevated concentrations of Hg and Cu. The maximum content of mercury in gold reaches 1.85 wt. %. At the same time, copper is most characteristic of the high-purity gold and can reach 0.24 wt. %. However, trace amounts of copper are sometimes found in silver-containing gold as well.

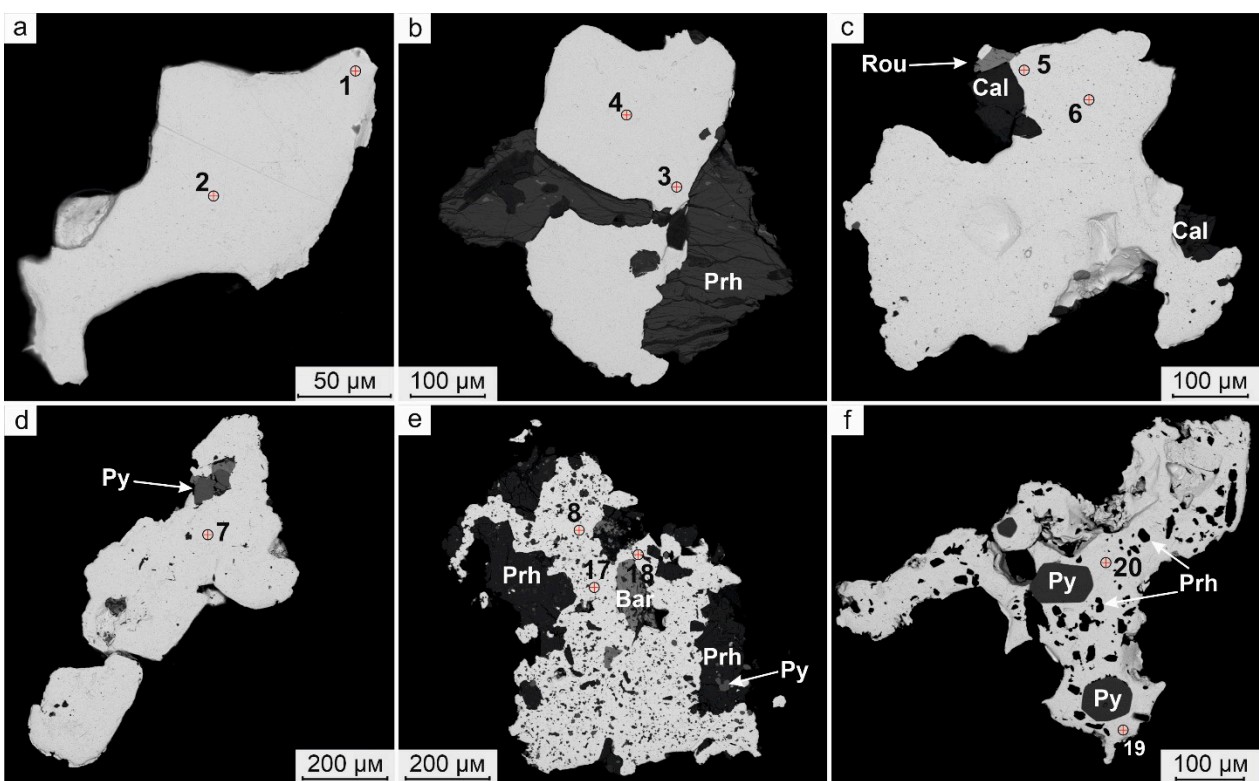

**Figure 10.** BSE-photo of polished sections of native gold grains. The analyses are given in the Table 3. (**a**) homogeneous gold grain; (**b**) native gold intergrows with prehnite (Prh); (**c**) native gold intergrows with calcite (Cal) and routhierite (Rou); (**d**) pyrite inclusion in native gold; (**e**) native gold with numerous inclusions of barite (Bar), prehnite (Prh) and pyrite; (**f**) euhedral inclusions of pyrite in native gold.

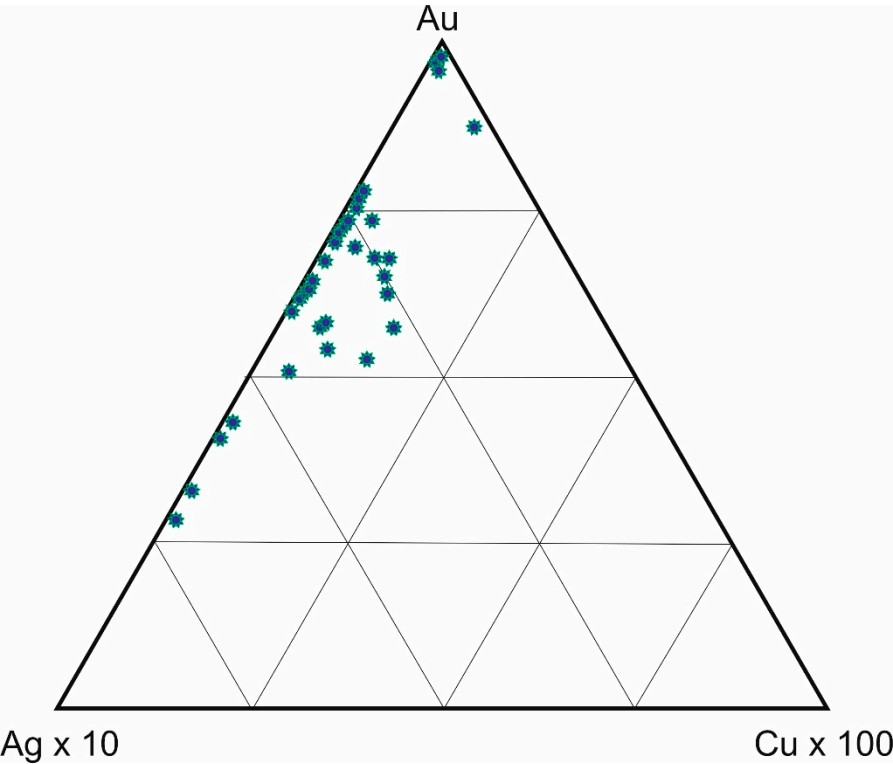

**Figure 11.** The Au–Ag × 10–Cu × 100 ternary diagram [19] for the native gold composition from ore-bearing breccias of the Vorontsovskoe deposit.

### 4.4. Chemical Composition of Native Gold–A Comparison

Admixtures of Ag, Cu and Hg are found in native gold from breccias with realgar-orpiment cement. The chemical composition of the native gold is similar to that obtained previously by other researchers [6,20]. Compared to the native gold from other ore types of the Vorontsovskoe deposit [6], the studied gold is featured by its higher fineness (>860) (Figure 12a). This characteristic, along with a uniform internal structure of the grains, distinguishes the native gold in breccias with realgar-orpiment cement from the gold from skarns and other type of gold mineralization found within the Vorontsovskoe deposit (see Figure 12a). The fineness of gold increases in the range from the most high-temperature ore associations (skarns) to the lowest-temperature breccias with realgar-orpiment cement. The last stages of development of the granite-related hydrothermal ore system associated with the Auerbakh intrusion are characterized by the formation of high-purity gold in a single paragenesis with Tl and Hg sulfosalts. It should also be noted that most of the gold compositional data in the classification diagram after [19] fall into the field of epithermal deposits.

High fineness of gold from realgar-orpiment breccias of the Vorontsovskoe deposit is a distinguishing feature in contrast to the native gold from other deposits in the Turyinsk-Auerbakh metallogenic province (Figure 12b), including copper-skarn and iron-skarn sub-economic deposits with low gold concentrations. Thus, the gold in the copper-skarn ores of the Bashmakovskoe and Bogoslovskoe deposits [21], which are located northwest of Auerbakh intrusive, has the lowest fineness due to the significant concentration of silver. A similar composition of native gold is also characteristic of the gold ore skarns in the Dorozhnoe ore occurrence found in 2016 [9], which is located between the Vorontsovskoe deposit and the Auerbakh intrusion. Low-fineness silver-containing gold is also characteristic of sub-economic ore deposits associated with quartz-sericite-altered tuffs within the Turyinsk-Auerbakh metallogenic province. Some similarities in the chemical composition of the studied native gold from the ore breccias of the Vorontsovskoe deposit have been established for a part of the analyses of gold from the iron ore skarns of the Yuzhno-

Peshchanskoe iron-skarn deposit. This similarity is likely a consequence of the unified ore formation process within the Vorontsovsko-Peshchanskaya ore-magmatic system [1].

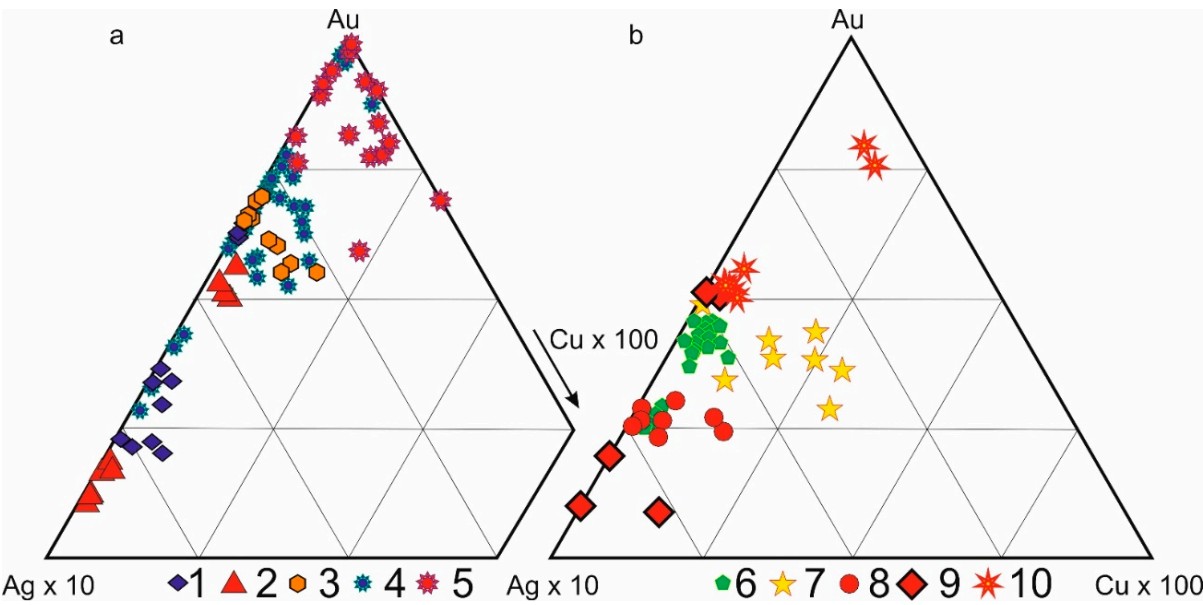

**Figure 12.** Ternary diagrams [19] for the native gold from different ore types in the Vorontsovskoe deposit (**a**) and other deposits in the Turyinsk-Auerbakh metallogenic province (**b**): 1—jasperoids [6]); 2—skarns [6]; 3—medium-temperature metasomatites developed on tuffs and tuffstones [6]; 4—gold from realgar-orpiment cement of ore breccia (this data); 5—gold from ore breccia [6]; 6—the skarns of the Dorozhnoe ore occurrence [9]; 7—the bornite-chalcopyrite exo- and endoskarns of the Bashmakovskoe deposit (19); 8—medium-temperature gold ore metasomatites on tuffstones (authors' data); 9—the chalcopyrite-pyrrhotite ores of the Bogoslovskoe deposit [21]; 10—the pyrite mineralization of the iron-skarn Yuzhno-Peshchanskoe deposit [21].

## 5. Discussion

The regular change in the composition of native gold from the most silver-enriched varieties to practically pure native gold in different ore types suggests an evolution of the ore-forming system associated with the Auerbakh polyphase intrusion As a result of the comparative analysis of the gold composition from different rocks of the Vorontsovskoye deposit—trend of changes in the composition of gold has been established. A similar trend with a decrease of silver in native gold depending on the temperature of metasomatite formation, has been established for the ore cluster as a whole [1]. The general geological and geochemical patterns of the Turyinsk-Auerbakh metallogenic province [1], including the presence of small non-economic porphyry copper deposits, suggest that the Vorontsovskoe deposit is an integral part of a large ore-magmatic system genetically associated with the formation of the Auerbakh intrusion. The composition of native gold from the ore-bearing breccias of the Vorontsovskoe deposit corresponds to epithermal deposits [18].

The significant amounts of Tl-containing ore minerals sharing the same paragenesis as native gold from breccias with realgar-orpiment cement in the Vorontsovskoe deposit also indicate a unique mineral gold ore assemblage that has no analogues among the deposits of the Urals. The presence of Tl-As-Hg-Sb-(Te) geochemical signature, including those belonging to the same paragenesis as gold, is also characteristic for Carlin-style gold deposits [22,23]. However, thallium minerals are also widely distributed in deposits associated with the metamorphosed carbonate strata composed of limestone and dolomite [24–26] with the formation of mineralized bodies according to the principle of alpine-type veins. However, in these deposits, the thallium mineralization is not accompanied by gold mineralization. Often, thallium minerals are also found in a distal disseminated Carlin-style deposit, such as the Allchar deposit [27–29] located in southern part of North Macedonia or

in deposits with intermediate position between epithermal deposits and sediment-hosted gold deposits (for example—the Jas Roux deposit, French Alps [30]).

In general, the presence of thallium mineralization in ore bodies of different genetic types can be explained by actively developed models of the magmatic formation of Carlin-style deposits [22–33]. Within these models, thallium mineralization can be localized at a maximum distance from the magmatic source in the zone of intermediate argillic alteration and the distance from the magmatic source can reach more than 5 km.

## 6. Conclusions

It is important to emphasize that the diversity of thallium minerals, including those first discovered at the Vorontsovskoe deposit—vorontsovite and ferrovorontsovite [34], tsygankoite [35], gladkovskyite [36], luboržákite [37], pokhodyashinite [38], gungerite [39] and auerbakhite [40]—suggest that the Vorontsovskoe deposit is a world-class mineralogical site.

Native gold in the ore breccias of the Vorontsovskoe deposit is concentrated in the form of both small grains ranging in size from 0.05 to 0.15 mm and fairly large grains up to 0.5 mm. In accordance with the technological classification of size [41], the gold of the Vorontsovskoe deposit belongs to the large type. The aggregates of native gold, considering their substantial distribution in the ore breccia cement, represent the bulk of the gold contained in the ores.

The high fineness of native gold from ore breccias distinguishes it from native gold found in other associations of the Vorontsovskoe deposit and other deposits of Turyinsk-Auerbakh metallogenic province. According to the composition, the native gold from the ore breccia cement in the Vorontsovskoe deposit differs from the native gold of other deposit ore types and from the other ore bodies in the Turyinsk-Auerbakh metallogenic province. The formation of gold in breccias occurred during the final stages of the formation of the Vorontsovsko-Peshchanskaya ore-magmatic system, together with barite, pyrite and Tl and Hg sulfosalts.

**Author Contributions:** Conceptualization, S.Y.S., R.S.P. and A.V.K.; Formal analysis, D.A.V., R.Š. and A.V.K.; Methodology, D.A.V. and R.Š.; Visualization, R.S.P.; Writing—review & editing, S.Y.S., R.S.P., D.V.K., L.N.S. and A.V.K. All authors have read and agreed to the published version of the manuscript.

**Funding:** This research was funded by the project of Institute Geology and Geochemistry UB RAS, state registration number AAAA-A18-118052590032-6.

**Acknowledgments:** We are grateful to Olga Plotinskaya for helpful suggestions that improved the manuscript and to Reviewers.

**Conflicts of Interest:** The authors declare no conflict of interest. The funders had no role in the design of the study; in the collection, analyses, or interpretation of data; in the writing of the manuscript, or in the decision to publish the results.

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
