# Peer review of "The Features of Native Gold in Ore-Bearing Breccias with Realgar-Orpiment Cement of the Vorontsovskoe Deposit (Northern Urals, Russia)"

_minerals, doi:10.3390/min11050541_

Round 1

Reviewer 1 Report

Review on the manuscript entitled “Gold-bearing fluid-explosive breccias of the Vorontsovskoe deposit (Northern Urals, Russia)” submitted by Sergey Yu. Stepanov, Roman S. Palamarchuk, Dmitry A. Varlamov, Darya V. Kiseleva, Ludmila N.

Sharpyonok, Radek Škoda and Anatoly V. Kasatkin for possible publication in Minerals

General comments

The manuscript submitted by Sergey Yu. Stepanov et al. deals with the native gold mineralization hosted by a peculiar ore-bearing brecciated horizon from the Vorontsovskoe gold deposit (Northern Urals, Russia). The striking feature of this gold-bearing mineralized structure relates to the nature of the cement which consists of a mineral assemblage dominated by realgar and orpiment. Based on morphological and chemical features of native gold and its relationship to other minerals, the authors propose that the Vorontsovskoe mineralization formed relatively far from the magmatic source of the Voronsovsko-Peshchanskaya ore-magmatic system, coincident with the emplacement of the Auerbakh gabbro-diorite-granite intrusion.

Overall, the paper is well written and the basic data presented by the authors could be of interest for the reader. My main concern with the paper relates to the discussion section which is highly speculative and is not based on robust evidences. One might wonder how we could propose a genetic model based only on the morphology and chemical compositions of gold. Too simplistic way of thinking. The authors should strengthen their model by cross-checking the information gathered in the course of their study with the complementary published investigations focusing on fluid inclusions to have an idea about the microthermometry and nature of the mineralizing fluids, alteration mineral assemblages and more importantly isotopic geochemistry to constrain the origin and significance of the hydrothermal system. Without doing so, the conclusions drawn will remain conflictual, unfounded, and more importantly speculative. Attached is my edit on the paper. I have made a few marginal comments as well.

In conclusion, the paper needs major revision particularly in regard of the discussion section.

Author Response

Dear Editor, dear Reviewers,

Thank you very much for the work done.

Your main remark is the speculative nature of the discussion regarding the comparison of the results obtained with the features of the Carlin type deposits. The other side of this issue is the determination of the genetic type of ores from the Vorontsovskoe deposit (Carlin type/epithermal type) based on the description of native gold.

In view of the coincidence of remarks from the editor and both reviewers, we decided to “lighten” the manuscript and almost completely removed the part with the comparative analysis of the morphological features and compositions of gold, as well as some geological regularities of the Vorontsovskoe deposit with various deposits around the World of the Carlin and epithermal types. Finally, the comparative analysis of gold compositions within the considered ore system has become the main result of the work. The comparison of the features of the Vorontsovskoe deposit with the deposits around the World has remained in general terms in the form of a comparative characteristic of the following parameters, i.e. is there thallium mineralization and gold mineralization at the site or not. No genetic interpretation is given proceeding from this.

We have tried to give detailed answers to the key questions of the editor and reviewers, which are given below. Minor notes on the text (grammar, etc.), especially from Reviewer 1, have been changed in the text without comments in editing mode. Figure 10 is added - SEM image of polished gold grains with analysis points shown below in Table 3.

Reviewer 2 Report

Conclusion Report

The mineralogical diversity of ore breccia is very interesting and possibly important in genetic reconstructions. However, some questions arise related to the argumentation and substantiation of decisions the authors came to. These are the General remarks. Other comments are of technical value and designated as minor comments.

General

  1. Abstract, Lines 25-28. Please, clarify this point: Do you mean far and independent or far but genetically related to the magmatic source?
  2. Results. What is the geochemical reason for such close association of elements Hg, Tl, As-Sb and S? Did initial rocks characterized with enhanced contents of these elements? Have you any attempt to compare these values? Why orpiment and realgar (the main constituents of breccia cement!) compositions were not studied? Such information is useful for the genetical purposes.
  3. Lines 267-270. Please, explain in more detail how the “morphological features” can characterize “stage of formation”.
  4. Lines 293-294. This conclusion is not motivated enough. For instance, Ag is abundant component of native gold but Ag minerals are absent in Table 2.
  5. Section 3.4. Line 326. Please, quantify, how much is the difference between “high-temperature” and “lowest-temperature” breccias.
  6. Lines 342-347. Please, specify “some similarities”, and explain why they are indicative of “unified ore formation process”.
  7. Discussion and Conclusions. Several questions arise in respect of the main conclusions of the manuscript (lines 442-445). Why the authors have not studied gold contents and speciation in the minerals they dealt with (including pyrite, arsenopyrite, realgar, etc.)? Why they exclude the possibility of Au exsolution from high-temperature ores contained Au in finely dispersed or structural form (typical for Carlin), and subsequently segregated and enlarged by aggregation into native gold particles? As indicated in ref.[7], the finely dissiminated Au-sulfide mineralization (including “invisible” gold) is typical for Vorontsovskoe deposit. Does it mean that ore breccias under study relate to another type of ore formation, different from that of the other parts of Vorontsovskoe deposit, which may belong to Carlin type? How it is possible? Moreover, if so, why this conclusion applies to the whole deposit?

Minor

Page 2, line 78. Unclear, what is (5 6).

Page 6.  What reference materials were used, natural or synthetic? Was FeS troilite? Please, give metrological details – precision, detection limits.

Page 11, line 277-278. Figure 9b, not 8b.

Page 13, Figure 10. Please, explain: field contours from [17] (?).  Points – the present work.

Page 16. Line 385. Something missed.

Line 409. In [40] no samples from “southern China” were mentioned, they were form Nevada, Papua New Guinea, Turkey, Fiji.

Author Response

(The authors gave the same response as above.)

Round 2

Reviewer 1 Report

The revised version is much better improved and desserves to be published. 

Well done.

Author Response

Thank you

Reviewer 2 Report

I'm satisfied  in principle with the corrections made by the authors in reply to my comments. However, ambiguity in interpretation is present as yet,  although  it does not exclude  publication in the newly presented form.

Author Response

Thank you.